

# Monitoring main ocean currents of the IBI region.

Álvaro de Pascual Collar[1], Roland Aznar[1], Bruno Levier[2], Marcos García Sotillo[1]

[1]Nologin Oceanic Weather Systems, Paseo de la Castellana 216, Floor 8th, Office 811, 28046 Madrid, Spain
[2]Mercator Ocean, 8-10, rue Hermès, 31520, Ramonville Saint-Agne, France

5  *Correspondence to*: Álvaro de Pascual Collar (a.depascual@nologin.es)

**Abstract.** The Iberia-Biscay-Ireland (IBI) region is located on the eastern margin of the North Atlantic. This geographical position results in a diverse array of currents, primarily including southward surface currents (associated to the North Atlantic subtropical gyre) and poleward slope currents at intermediate depths following the continental margins of Africa and Europa. Ocean currents have significant climatic, environmental, economic, and social implications, making them a crucial parameter whose variability needs to be monitored to anticipate diagnosis and support decision-making in the face of changing scenarios.

The present study proposes a methodology that allows for systematic monitoring of ocean currents. This methodology is based on calculating volume transports within monitoring windows defined using: (i) present knowledge of the ocean and (ii) delineation of the water mass to be monitored based on its density range. The proposed indicator is computed using various sources of observational and modeling data, resulting in a multiproduct output. This approach provides not only an Ocean Monitoring Indicator (OMI) of transport anomalies but also an analysis of uncertainties.

The calculation of this OMI on the currents in the IBI region shows that, despite the high uncertainties, the index is capable of detecting events of high/low transport intensity as well as significant transport trends superimposed on the interannual variability of some of the analyzed currents.

## 1 Introduction

The IBI (Iberia-Biscay-Ireland) region encompasses the Northeastern Atlantic Ocean from the Canary Islands (latitude 28ºN) to the coasts of Ireland and Great Britain (latitude 60ºN). This zone is defined as one of the Marine Forecasting Centres of Copernicus Marine Service (Figure 1). The IBI area is a very complex region characterized by a remarkable variety of ocean processes and scales (Sotillo et al., 2015). The western, and deeper, side of the IBI domain is affected by large-scale currents, mainly the closure of the North Atlantic Drift, where it splits into two branches, the major one continuing north along the northwestern European shelves (Bower et al., 2002; Holliday et al., 2008) and the other, goes eastward to form the eastern boundary current of the North Atlantic subtropical gyre. This boundary current is composed of the Azores Current (Jia, 2000; Peliz et al., 2007), the Portugal equatorward current (Pérez et al., 2001) and the Canary Current (Knoll et al., 2002; Mason et al., 2011). Along the European continental slope, a subsurface current flow northward following the coasts of Portugal, Spain (Mazé et al., 1997) and the Celtic-Armorican Slope (Fricourt et al., 2007). The signature of this current can




be observed as far north as Porcupine Bank (52ºN) (White & Bowyer, 1997, Fricourt et al., 2007, Pascual et al. 2018, de Pascual-Collar et al., 2019). The intermediate levels of the Canary basin are affected by the northward-flowing Antarctic Intermediate Water. The flow of this globally important water mass has been described in the Lanzarote passage, located between the Canary Islands and the African coast (Knoll et al., 2002; Machin et al., 2010). The Alboran Sea, located in the

Western Mediterranean, is part of the IBI domain. There, the Atlantic surface inflow through the Gibraltar Strait causes an eastward flow of modified Atlantic Water, forming the Western Mediterranean Water. This eastward flow is constrained near the African coast after passing the Almería-Oran front (Tintoré et al., 1988; Benzohra and Millot, 1995; Font et al., 1998), creating the Algerian Current (Font et al., 1998; Sotillo et al., 2016).

The variability of ocean currents in the IBI domain is relevant to the global thermohaline circulation and other climatic and

environmental issues. For example, as discussed by Fasullo and Trenberth (2008), subtropical gyres play a crucial role in the meridional energy balance. The poleward salt transport of Mediterranean water, driven by subsurface slope currents, has significant implications for salinity anomalies in the Rockall Trough and the Nordic Seas, as studied by Holliday (2003), Holliday et al. (2008), and Bozec et al. (2011). The Algerian current serves as the sole pathway for Atlantic Water to reach the Western Mediterranean. Furthermore, the mixing processes between the Antarctic Intermediate Water and the

Mediterranean Water in the African continental slope influences the salinity and nutrients concentration of the Mediterranean Water as it spreads into the North Atlantic (van Aken, 2000; Machin et al., 2010; de Pascual-Collar, 2019). Additionally, ocean currents also influence the spreading of biological populations (Daewel et al., 2008)

One of the main challenges in monitoring ocean currents is the lack of observational data for subsurface transport. A

significant portion of oceanic transports occur in subsurface levels where observational data is scarce. In this regard, modeling data provides an alternative as it offers physically consistent results in those levels where observations are lacking. Additionally, since modeling data is gridded, it allows for the approximate calculation of oceanic water mass transport. However, it is worth noting that modeling results at these intermediate and deep levels have a limited level of validation. To address this issue, ensemble approaches can be employed to estimate the uncertainties of the results, thereby improving the

reliability of the findings.

In addition to climate-related regional applications, such long-term monitoring of currents in the IBI region can be of great interest for those ocean-related activities, for which such information on regional currents variability can enhance decision-making processes in the blue economy sectors (Rayner et al., 2019). Particularly useful for maritime navigation and safety (Rayner et al., 2019), fight against pollution, fisheries, aquaculture, marine renewable energy (Cabagnaro et al., 2020) and

scientific research. By understanding the dynamics of ocean currents, stakeholders in these sectors would be able to optimize their operations, improving safety, and planing/developing more sustainable practices. The availability of accurate and up-to-date information on regional ocean currents would contribute to the overall efficiency and effectiveness of ocean-related activities in the IBI region.



The aim of this work is to provide a pool of multi-product OMIs (Ocean Monitoring Indicators) focused on monitoring the activity of the main currents described in the Iberia-Biscay-Ireland regional seas. To this aim, several monitoring vertical sections are proposed, and anomalies of transverse water transports are computed, providing information on the variability and trends of the monitored regional currents.

## 2 Data and methods

The present study proposes a methodology to monitor the main ocean currents in the IBI region. This is achieved by analyzing the velocity field in the region and proposing representative monitoring windows where the volume transport is monitored. Since the proposed monitoring windows are defined to follow specific water masses, they are defined attending not only to the spatial representativeness of the region but also defining a range of specific densities representative of the targeted water mass.


In order to provide an ensemble-based final result, five Copernicus products listed and referenced in Table 1 have been used. Among these products, there are four reanalysis products (GLO-REA, IBI-REA, NWS-REA, and MED-REA) and one product obtained from reprocessed observations (GLO-ARM).

To perform monitoring of the currents described in the introduction, some vertical sections have been defined (see

geographical locations in Figure 1 and further details in Table 2). These sections have been proposed based on the existing bibliographic description of the main currents occurring in the IBI region. Therefore, the RT (Rockall Trough) section has been defined to monitor zonal transports of Mediterranean-origin water towards the North Sea (Holliday et al., 2008; Lozier and Stewart, 2009). The N48 Section has been included to monitor total transports along the parallel 48ºN; this section corresponds to another OMI operationally delivered by the Copernicus Marine Service (EU Copernicus Marine Service

Product, 2019) but computed in the IBI domain. The CAS (Celtic-Armorican Slope) Section has been defined to monitor northward flows along the European continental shelf (Fricourt et al., 2007). The WIP (West Iberian Peninsula) section aims to study both the northwards flow of intermediate Mediterranean water known as Iberian Poleward Current (Daniault et al., 1994; Mazé et al., 1997; Fricourt et al., 2007) and the southwards wind-induced surface flows along Portuguese coast also known as Portugal Current (Pérez et al., 2001). The ABB (Algerian-Balearic Basin) Section has been defined to analyze the

behaviour of the Algerian Current (Tintoré et al, 1988; Benzohra and Millot, 1995; Font et al., 1998). The AS (African Slope) section has been defined to monitor surface transports induced by the trade winds along the coasts of the African continent, commonly known as the Canary Current (Knoll et al. 2002, Mason et al. 2011). Similarly, the MA (Madeira) Section is intended to monitor westward zonal surface flows near the island of Madeira, commonly referred to as the Azores Current (Mason et al., 2011). Finally, the LP (Lanzarote Passage) Section has been defined to monitor the flow of deep

Antarctic water between the African continental shelf and the Canary Islands (Knoll et al., 2002; Machin et al., 2010).



The OMI defined in this study aims to monitor currents, by calculating volume transport anomalies in specific layers inside the previously described vertical sections. Since all the products used are in a gridded format, the vector field of transports is computed by multiplying the velocity field components by the cross-sectional area of the calculation grid cells for each product. As each product is provided on a particular grid with a specific resolution, this step is performed using the grid mesh of each product.

Within each section (i.e., RT, CAS, WIP, AS, MA, LP, ABB, and 48N), the signal of each desired flow associated to specific water masses to be monitored have been identified. This identification is based on the oceanographic description of the region obtained from the literature. Figure 2 presents the results of transverse velocity obtained in the vertical sections defined in Figure 1. These sections are based on the literature and were described in the Introduction and Data and Methods sections of this study. The meridional section defined in Rockall Trough (RT) shows two areas presenting intense and opposite zonal transports: the southern area (located between ~53 ºN and ~54ºN) marked by the eastward transport of modified Mediterranean Water into the Nordic Seas, and the northern side, located between latitudes ~55.5 ºN and ~56 ºN, showing a subsurface westward transport leaving Rockall Trough. The zonal section defined in the Celtic-Armorican Slope (CAS) depicts a subsurface poleward flow attached to the continental slope, with maximum intensity of the current located at 1000 m depth. However, the northward flow affects the complete water column, from few hundreds of meters to the bottom. The zonal section proposed in the Western Iberian Peninsula (WIP) displays a subsurface poleward flow with maximum speed close to the continental slope and a surface equatorward current affecting waters over the continental platform of Portugal. The zonal section defined in the African Slope (AS) displays a similar structure than the section in the Western Iberian Peninsula, marked by an equatorward surface flow over the continental shelf, corresponding to the so-called Canary Current and an opposite subsurface northward flow attached to the continental slope. The meridional section north of Madeira (MA) has been defined to monitor the so-called Azores Current, this current is reflected as an eastward transport seen in upper layers (up to 600 m depth) being the center part of the current influenced (and almost split) by the presence of a sea mountain north of Azores and below 600 m depth transports are strongly influenced by bathymetry. The zonal section defined in the Lanzarote Passage (LP) exhibits a clear poleward transport close to the continental slope, this transport is split into two levels; the upper-intermediate level comprises from 200 m up to 600 m depth and corresponds to the slope poleward current seen in other sections (such as the African Slope, Western Iberian Peninsula and Celtic-Armorican Slope). The intermediate-deeper level of the poleward current seen in the Lanzarote Passage section comprises the waters from 500 m depth up to the bottom and it is associated with the flows of Antarctic Intermediate Water.

The consistency of the mean transverse velocity in vertical sections found in Figure 1 and the described currents in literature provide the support to propose the lateral boundaries of the monitoring windows defined in Table 2 and marked with vertical dashed green lines in Figure 2: Rockall Trough Eastward (RTE), Rockall Trough Westward (RTW), Armorican Slope Poleward (ASP), Iberian slope Poleward (IBP), Portugal Current (PC), Azores Current (AZ), Canary Current (CC), and





Antarctic Water (AW). It is worth mentioning that no monitoring windows have been included in the N48 section since the generated OMI will compute the total transport along the 48 ºN parallel.

Since the tracked currents involved specific water masses, it is considered that the vertical boundaries of the monitoring windows exhibit vertical variability. Thus, the vertical boundaries of the monitoring windows have been defined based on the upper and lower density layers characterizing each monitored water mass. Given that density field exhibits temporal variability within each product, the vertical boundaries of each window are dynamic, adapting to the oscillations in the ocean's density field. To find these vertical limits, the T/S diagrams have been calculated for each monitoring window, temporally averaged over the period 1993-2021. Figure 3 shows the calculated T/S diagrams for each monitoring window using the IBI-REA product. The solid density lines marked on each diagram indicate the boundaries that have been selected to define the water mass characterizing each current. Additionally, these density boundaries are also represented in Figure 2 as horizontal dashed lines in green. Since there are some monitoring windows whose vertical boundaries may be defined by the surface (PC, CC, AC, and ALC windows) or the seafloor (AW window), some panels in Figure 3 show a single density line. Table 2 presents the density-defined vertical boundaries used for each window, including those windows with surface or seafloor as their vertical limit.

Subsequently, the transport estimates are integrated within each defined monitoring window monthly basis. As explained earlier, the vertical boundaries of the monitoring windows are dynamic and defined based on specific density layers. The UNESCO equations (Fofonoff and Millard, 1983) are used to estimate the density field from the salinity and potential temperature fields of each product. Thus, each monitoring window is defined at each time step with fixed lateral boundaries (defined as static latitude/longitude locations) and variable vertical boundaries that may vary in time according to vertical oscillations of the density field (Table 2).

After spatial integration, estimated transport timeseries is obtained for each product and each monitoring window. Since the proposed OMI aims to focus on interannual and longer time scales, seasonal variability is filtered out, being the data processed as anomalies by removing monthly mean from each value, using the climatological average of the time series in the period 1993-2021 as reference.

Finally, outcomes from the different products are combined to produce an ensemble, being showed the mean of all products and the standard deviation. Since not all Copernicus products used in this study cover the entire IBI region, some products cannot be used for current monitoring in certain monitoring windows (Table 2 provides a list of the products used to monitor transports in each Section).



# 3 Results and discussion

The anomaly of volume transports in specific monitoring windows (relative to the monthly mean transport between 1993 and
2021) is proposed as OMI for each regional current (Figure 4). The uncertainty is assessed by the estimation of indicators
with several different Copernicus Marine products (OMIs are estimated with the more complete set of products available in
each monitoring window; see Table 2); the higher is the agreement of results provided by diverse systems the higher is its
statistical robustness.

The interpretation of the OMI results presented in Figure 4 should be done considering that the calculated average transports
include the sign of the current velocity vector (positive for northward and eastward transports, and vice versa). Therefore, the
anomalies shown in the OMI should be interpreted following the same sign convention. Thus, a positive anomaly can
indicate an intensification, weakening, or even a reversal of the current. Among the proposed monitoring windows, RTE,
ASP, IBP, AW, and ALC exhibit a positive mean velocity, so positive transport anomalies imply an intensification of the
current. On the other hand, the RTW, PC, CC, AC, and N48 monitoring windows show negative mean transport values,
where intensification of the current is represented by negative transport anomalies.

Analyzing the uncertainties of the time series, can be observed that differences among the products sometimes hinder the
detection of significant anomalies. This is the case for the ASP, IBP, PC, and AW currents, for which anomalies exceeding
the data dispersion are scarce. Conversely, there are other monitoring windows, where a lower data dispersion allows for a
more precise estimation of the overall variability. This is the case for the CC and ALC currents, where, due to being more
stable and shallower currents, the uncertainties are generally smaller than the magnitude of the anomalies. In the remaining
monitoring windows (RTE, RTW, AC, and 48N), uncertainties are considerable but sufficiently low allowing identification
of specific events with pronounced variability. In this regard, it can be highlighted as examples the events of negative
anomalies (1995-1996, 1999) and positive anomalies (2010 and 2014) in the RTE section, the positive anomaly of the
Azores Current (AC) in 2006, or the anomalous transports along the N48 section in 2003 and 2012.

Analyzing the trends in the OMIs, Figure 4 shows the absence of statistically significant trends in most of the monitoring
sections. However, the sections related to the Mediterranean water flow along the continental shelf do exhibit opposite
trends, where the ASP window indicates a decrease of 0.02 Sv/Yr in northward flow along the Celtic-Armorican Slope and
an intensification of transport towards the Scandinavian Seas of 0.01 Sv/Yr.

# 4 Conclusions

This study proposes a methodology for defining OMIs to monitor variability of the main ocean currents in the IBI region.
The objective is to describe through the proposed indicator strengths and weaknesses of the transports associated. The
methodology used to define the OMI involves utilizing various Copernicus Marine products, which allows for an ensemble
analysis that not only provides information on the currents, but also yields an assessment of uncertainties associated.



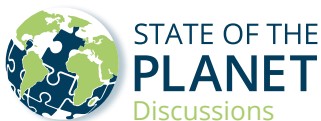

The proposed indicator is based on the definition of monitoring spatial windows that are representative of the overall
transport of the current. These spatial windows are defined based on two criteria: Firstly, horizontal boundaries determined
using initially bibliographic criteria and by detection of the targeted currents in the averaged velocity fields. Secondly,
vertical limits defined based on the observed density of water masses, allowing this methodology, vertical limits of the
monitoring window not statically defined, but adaptative to the inherent vertical variability of water masses.

The results have demonstrated that the OMI is capable of monitoring the targeted regional currents with varying degrees of
uncertainty. The different products used show a higher agreement in shallower and more persistent currents, while the
indicator exhibits more uncertainties when monitoring deep currents or those with higher temporal variability. Nevertheless,
and despite the magnitude of the uncertainties, the OMI is able to detect periods of high or low activity in six of the proposed
monitoring windows (RTE, RTW, CC, AC, ALC, and N48). Outstanding events, including periods of low activity in the
Canary Current (in 1996 and 2010), and in the Azores Current, in 2006, as well as a significant intensification of transports
associated to this Azores Current, observed in 2021.

The OMI also allows detection of statistically significant trends in some of the monitoring spatial windows. Specifically, the
RTE and ASP windows show opposite trends in water transports. While there is a transport increase towards the Nordic Seas
in the Rockall Trough, and a decrease in Mediterranean water transported northward through the Celtic-Armorican
continental slope.

Finally, this study highlights the utility of model data products for monitoring ocean currents. This methodology is
particularly relevant for tracking deep currents where observational data availability is limited. The proposed methodology
may have a further application in a variety of contexts, such as maritime navigation, fishing, aquaculture, marine renewable
energy, and general scientific research. Indeed, next steps in these fields will involve expanding the methodology to monitor
not only water transports, but the transports of other properties, such as ocean heat transport, freshwater/salt transports, or
other biogeochemical properties. enhancing the interest of such OMIs on specific fields, such as the study of planetary
climate variability.

**Competing interests**

The contact author has declared that none of the authors has any competing interests.

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

| Product ref. no. | Product ID<br>Acronym<br>Type | Data access | Documentation:<br>**QUID:** Quality Information Document. |
|---|---|---|---|



| | | | **PUM:** Product User Manual. |
|---|---|---|---|
| 1 | GLOBAL_MULTIYEAR_PHY_001_030 (GLO-REA) Numerical models (reanalysis) | EU Copernicus Marine Service Product (2022a) | QUID: Drévillon et al. (2022a) PUM: Drévillon et al. (2022b) |
| 2 | IBI_MULTIYEAR_PHY_005_002 (IBI-REA) Numerical models (reanalysis) | EU Copernicus Marine Service Product (2022b) | QUID: Levier et al. (2022) PUM: Amo-Baladrón et al. (2022) |
| 3 | MULTIOBS_GLO_PHY_TSUV_3D_MYNRT_015_012 (GLO-ARM) Reprocessed observations | EU Copernicus Marine Service Product (2021a) | QUID: Greiner et al. (2021) PUM: Guinehut (2021) |
| 4 | NWSHELF_MULTIYEAR_PHY_004_009 (NWS-REA) Numerical models (reanalysis) | EU Copernicus Marine Service Product (2021b) | QUID: Renshaw et al. (2021) PUM: Tonani et al. (2022) |
| 5 | MEDSEA_MULTIYEAR_PHY_006_004 (MED-REA) Numerical models (reanalysis) | EU Copernicus Marine Service Product (2022c) | QUID: Escudier et al. (2022) PUM: Lecci et al. (2022) |

**Table 1: List of Copernicus Marine products used for the computation of currents in Iberia-Biscay-Ireland region (IBI).**




| Section (Figure 1) | Monitoring Window (Figure 2) | Latitude | Longitude | Density (gr/m3) | CMEMS products |
|---|---|---|---|---|---|
| Rockall Trough (RT) | Rockall Trough Westward (RTW) | 55.5 / 56.0 | -14.7 | 27.32 / 27.7 | GLO-REA IBI-REA GLO-ARM NWS-REA |
| | Rockall Trough Eastward (RTE) | 52.7 / 53.8 | -14.7 | 27.25 / 27.65 | |
| Celtic-Armorican Slope (CAS) | Armorican Slope Poleward (ASP) | 46.0 | -5.0 / -4.1 | 27.25 / 27.8 | |
| West Iberian Peninsula (WIP) | Iberian Poleward (IBP) | 42.7 | -10.35 / -9.6 | 27.1 / 27.75 | |
| | Portugal Current (PC) | 42.7 | -9.6 / -9.0 | Sfc / 26.9 | |
| Algerian-Balearic Basin (ABB) | Algerian Current (ALC) | 36.6 / 37.3 | 1.5 | Sfc / 27.25 | GLO-REA IBI-REA GLO-ARM MED-REA |
| Madeira (MA) | Azores Current (AC) | 32.9 / 36.5 | -16.9 | Sfc / 27.25 | GLO-REA IBI-REA GLO-ARM |
| African Slope (AS) | Canary Current (CC) | 33.0 | -9.5 / -8.8 | Sfc / 26.7 | |
| Lanzarote Passage (LP) | Antarctic Water (AW) | 29.0 | -13 / -12.4 | 27.1 / Sfl | |

**Table 2: Description of the monitoring windows used to compute the ocean current OMI. Min/max values of latitude, longitude and density anomaly are detailed (sfc used for surface, sfl for seafloor).**








**Figure 1: Bathymetry of the IBI domain. Dotted red lines indicate the location and acronyms of the sections used to calculate the mean transverse velocity: RT (Rockall Trough), 48N (48 degrees North), CAS (Celtic-Armorican Slope), WIP (West Iberian Peninsula), ABB (Algerian-Balearic Basin), AS (African Slope), MA (Madeira), and LP (Lanzarote Passage). Arrows represent the currents monitored in each section, categorized by an arbitrary color scheme: Blue arrows indicate upper levels (0-500 meters), green arrows indicate intermediate depths (500-1500 meters), and red arrows indicate deep levels (>1500 meters).**




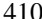

**Figure 2: Vertical sections of transverse velocity along the transects defined in Figure 1. Green dashed lines correspond to the monitoring windows defined to describe the variability of the currents defined. RTW: Rockall Trough Westward, RTE: Rockall**
**Trough Eastward, ASP: Armorican Slope Poleward, IBP: Iberian Poleward, PC: Portugal Current, ALC: Algerian Current, CC: Canary Current, AC: Azores Current and AW: Antarctic Water.**



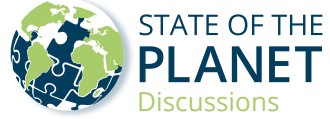

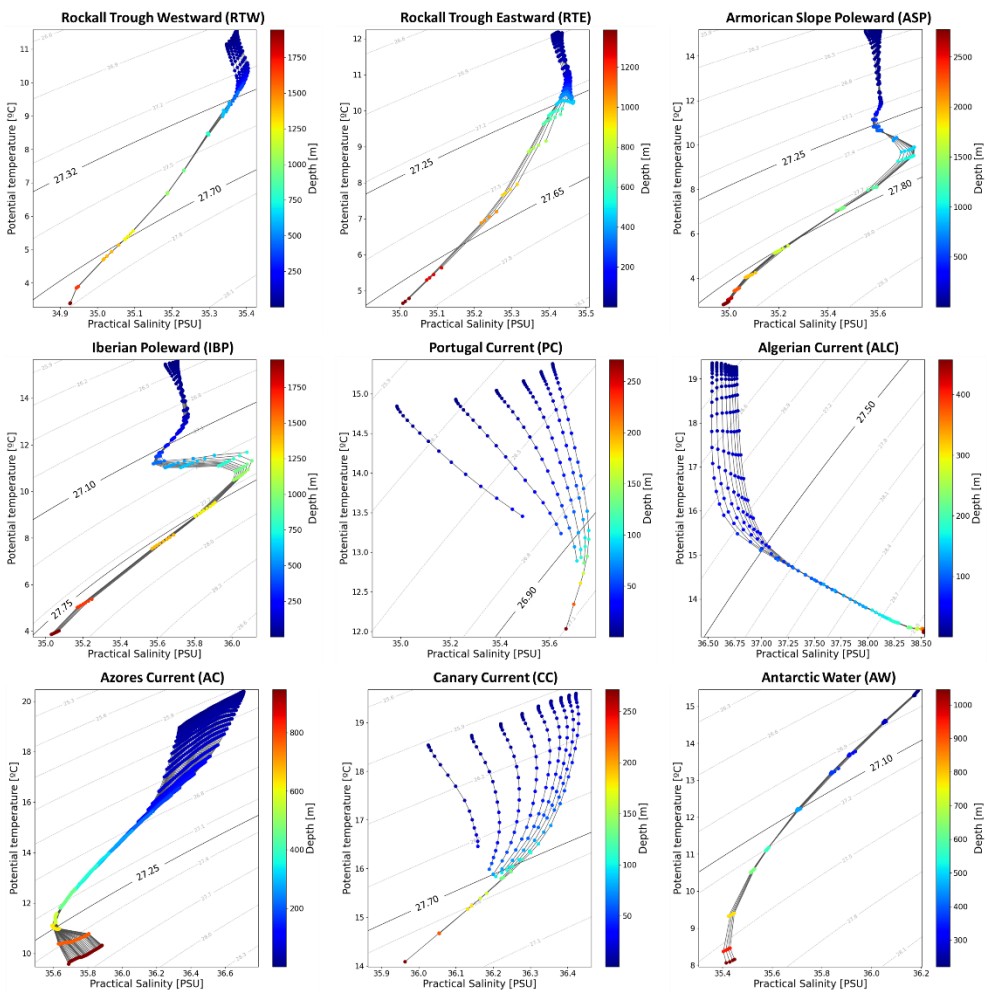

**Figure 3: Averaged θ/S diagrams computed on the monitoring sections using IBI-MY product data over the period 1993-2021. The diagrams include the anomaly of potential density field (shown as dotted grey lines), being the vertical boundaries of each monitoring section represented by the black continuous lines. The diagrams that have a single continuous line correspond to monitoring windows where one of their vertical levels is defined as surface or seafloor (see Table 2).**





**Figure 4: Annual anomalies of cross-section volume transport averaged in monitoring windows RTE, RTW, ASP, IBP, PC, CC, AC, AW, ALC, and N48. Time series computed and averaged from different Copernicus Marine products for each window (see Table 2) providing a multi-product result. The blue line represents the ensemble mean, and shaded grey areas represent the standard deviation of the ensemble. The analysis of trends (at 95% confidence interval) computed in the period 1993–2021 is included (bottom right box). Trend lines (dashed line) are only included in the figures when a significant trend is obtained. Arrows indicate the direction of the mean flow through the sections.**