# Peer review of "Monitoring main ocean currents of the IBI region."

_State of the Planet, 2023_

## Author Comment (AC1)

**General comments**

The manuscript deals with the complex surface and deep ocean circulation in the Iberian-Biscay-Ireland (IBI) region of the Northeast Atlantic Ocean, highlighting the importance of examining the currents for various applications such as climate research, marine navigation, fisheries and marine renewable energy. The study proposes a methodology for the creation of ocean monitoring indicators (OMI) to assess the interannual variability of the main ocean currents in this region.

The methodology involves defining specific monitoring windows based on existing literature, dynamic vertical density boundaries, and using Copernicus Marine products and ensemble methods to calculate annual volume transport anomalies within these windows.

The results show the variability of regional currents, with positive anomalies indicating either intensification or weakening of currents. Despite the uncertainties, some OMIs successfully detect periods of high or low current activity, and reveal significant trends.

The work is clearly presented and illustrated. As far as I know, this substantial work appears to take into account the main ocean currents in the region, and the whole seems worth publishing. However, a number of questions remain concerning the definition of the boundaries of certain sections, and these need to be clarified. In particular, it would be essential to specify the extent to which transport uncertainties for certain sections are linked to the choice of vertical and horizontal boundaries.

**Specific comments**

SC1 → Figure 2: To which reanalysis (es) and for which period do the current fields shown in this figure correspond?

> The figure caption will be modified including this information.

SC2 → Figure 2 and table 2: The upper isopycnal limits chosen correspond roughly to the main thermocline. Have you tested the sensitivity of transport to the choice of this isopycnal value?

> Yes, we have conducted sensitivity analysis of the boundaries for all monitoring windows (both vertical and horizontal).

The analysis revealed that the correlation of the results is high when modifying each of the boundaries, particularly when the modifications entail small variations in the total surface area of the monitoring window. Thus, for all boundaries of the monitoring windows, it is observed that small changes in the parameters defining the window (latitude, longitude, or density) result in equally small changes in transport anomalies.

This is because the time series presented in Figure 4 are expressed as anomalies; therefore, changes in the time series are only noticeable when the monitoring window is modified to include other oceanographic processes distinct from the current intended for monitoring.

This observation also applies to the selection of density boundaries in cases where it marks the thermocline. Modifying the chosen isopycnal value to delineate this boundary constitutes a small percentage of the total surface area of the monitoring window.

In the following figure we show an example that arise as a result of a suggestion of the other referee of the work:

[Figure]

On this example the result of modifying the lower boundary of the monitoring window ALC can be observed. The isopicnal limit was modified

from 27.5 gr/m3 (panels on the left column) to 28.5 gr/m3 (panels on the right column). This alteration results in a lowering of the lower limit of the monitoring window by approximately 100 m (panels in the upper row), thereby enhancing the capture of the total transport of the Algerian Current. Additionally, it is observed that the lower limit shifts from being adjacent to the main thermocline to being positioned below it (T/S diagrams in the second row).

As evident in the OMI plots presented in the third row, the outcome of this modification is practically indistinguishable. The time series exhibit a change in the obtained mean transport (0.32 Sv in the left case and 0.50 Sv in the right case). However, since the time series are presented in terms of anomalies, a modification in absolute variability is noted (anomaly values are higher in the right panel), but uncertainties, trends, and notable events remain unaltered.

SC3 → How was the outer (offshore) boundary of the different currents defined? The geographical limits seem rather arbitrary. Do they result solely from information based on the literature, or – which would seem more appropriate here – from an analysis of the variability of current intensity or transport for the different reanalyses? This is the case, in particular, for the Celtic-Armorican Slope section and the Armorican Slope Poleward current.

The very definition of monitoring windows entails an arbitrary component since the same current can be monitored using very different reference sections. For instance, the Armorican Slope Poleward current could be monitored on the continental slope at different latitudes ranging from 45ºN to 47ºN.

The selection of the region where we have chosen to monitor each current has been made based on the literature, attempting to locate the monitoring window in areas where the majority of studies describing it are concentrated and where the data we are working with seem to depict it more clearly.

However, once the region for monitoring each current has been selected, the definition of latitude and longitude values, as well as the density limits defining each current, has been established based on the mean values of the data.

SC4 → Figure 3: What do the different traces (lines) in the $\theta$-S diagrams represent? Monthly climatological average? Results from one or more reanalyses? Please clarify.

The figure caption will be modified including this information.

SC5 → Discussion and Figure 4. Just out of curiosity, regarding the interpretation of interannual variability, is it by any chance the case that some of the significant signals you have highlighted have also been listed in the literature?

According to our investigations, we haven't found many works focused on the transports of specific currents. This is generally because the works we come across typically concentrate on calculating transports along complete sections without focusing on specific water masses.

Additionally, the comparison of transports with the literature becomes very complex since each transport must be calculated in a section with specific values of latitude, longitude, and depth. Therefore, the mean values are conditioned by the dimensions of the section.

We found the work of Houpert et al. (2020) particularly interesting, where they conduct a study of transports through a zonal section in Rockall Trough at 57.4ºN. We would have liked to define the RT section at this location, but unfortunately, the meridional limit of the IBI product is at 56ºN, making it unfeasible to calculate transports at latitudes so far north.

However, we have identified some bibliographic works (New and Smythe-Wright, 2001) that allow for comparisons of the mean current speed. We propose introducing these types of comparisons in a future version of the manuscript.

Houpert, L., Cunningham, S., Fraser, N., Johnson, C., Holliday, N. P., Jones, S., ... & Rayner, D. (2020). Observed variability of the North Atlantic Current in the Rockall Trough from 4 years of mooring measurements. Journal of Geophysical Research: Oceans, 125(10), e2020JC016403. https://doi.org/10.1029/2020JC016403.

New, A. L., and Smythe-Wright, D. Aspects of the circulation in the Rockall Trough. Continental Shelf Research, 21(8-10), 777-810, https://doi.org/10.1016/S0278-4343(00)00113-8, 2001.

**Technical corrections**

TC1 → Line 31 : change « Fricourt et al., 2007 » by « Friocourt et al., 2007 ».

Accepted

TC2 → Figure 2: The AC current on the Madeira section is barely legible. The contrast should be increased.

The figure will be modified to enhance the readability of the acronyms.

TC3 → The following reference in the bibliography is not cited in the text :
Cavagnaro, R. J., Copping, A. E., Green, R., Greene, D., Jenne, S., Rose, D., Overhus, D.: Powering the blue economy: Progress exploring marine renewable energy integration with ocean observations. Marine Technology Society Journal, 54(6), 114-125, https://doi.org/10.4031/MTSJ.54.6.11, 2020.

This reference will be removed from bibliography.

---

## Author Comment (AC2)

Review of "**Monitoring main ocean currents of the IBI region**"

By Á. de Pascual Collar, R. Aznar, B. Levier, M. García Sotillo.

The authors propose a methodology to monitor several surface and subsurface ocean current transports of the Iberia-Biscay-Ireland region, which allows to investigate their interannual/decadal variability or long-term trends.

Defining the lateral (longitude/latitude) and vertical boundaries (density range) of specific ocean currents is a tricky task. The choice of those boundaries might - sometimes often - lead to significant uncertainties. The uncertainties are assessed by comparing a set of five Copernicus products.

The methodology is clear, the figures adequate, and the manuscript well written. The manuscript is worth publishing after the minor changes listed below.

**Main comments:**

- I would provide the mean transport values in the results section (even the seasonal variability). Figure 4 shows the annual anomalies. It is however useful to know the mean values. It is very different to have a 1Sv anomaly for a 1Sv mean current (100%) or a 1Sv anomaly for a 10Sv mean current (10%). It is ok to keep the y-axis in Sv and not in % of the mean value.

We found this suggestion to be highly valuable. Following this advice, we have devised a method to incorporate the mean transports into Figure 4. This has led to the development of a new paragraph in the discussion section, wherein we present additional findings. These include a discussion on the statistical significance of results in monitoring windows N48 and PC, as well as the identification of occasional halts and reversals in transport in other monitoring windows.

However, due to the high uncertainties found, we finally decided not to include the information of seasonal variabilities. We consider this result would be really interesting, but it should be preceded by some kind of comparison of results between the different data sources involved. Additionally, it should be noted that this work utilizes some globally oriented databases that may encounter challenges in representing the seasonal variability of certain currents with more localized scales as presented in this study.

- Either in the Introduction or Results sections, it would be interesting to compare the mean values of these transports to estimated values based on in-situ data from the literature.

The comparison of results with other studies is challenging. This difficulty arises because, while we focus our attention on widely referenced areas and currents, the small calculation specifications make comparisons unreliable. There is some room for comparing results in terms of average velocity with other studies; however, the calculation of transports must be performed by defining a section with specific latitude, longitude, and depths. This condition influences the obtained mean transport, significantly limiting the identification of studies with directly comparable results.

We conducted a literature search to find results that could be comparable to the present study. Below, we provide a brief discussion of the studies that appeared to be more comparable.

The work of Houpert et al. (2020) presents an analysis of transports in Rockall Trough. It would be of great interest to compare the results of this work with Houpert et al. (2020). However, the calculation section in this work is situated around 57.4ºN. Unfortunately, the meridional limit of the IBI model is at 56ºN, making it unfeasible to calculate transports at latitudes so far north.

Several of the identified works specifically focus on the analysis of the seasonal variability of currents (e.g., Fricourt et al., 2007, or Tales-Machado et al., 2016). While these works allow for the comparison of current speeds, focusing on seasonal values provides limited insights beyond confirming that the orders of magnitude are consistent.

Nevertheless, we have found some works that can be discussed in the contribution, such as New and Smythe-Wright (2001). Therefore, we propose including in a new version of the manuscript some comparisons with existing studies.

Houpert, L., Cunningham, S., Fraser, N., Johnson, C., Holliday, N. P., Jones, S., ... & Rayner, D. (2020). Observed variability of the North Atlantic Current in the Rockall Trough from 4 years of mooring measurements. Journal of Geophysical Research: Oceans, 125(10), e2020JC016403. https://doi.org/10.1029/2020JC016403.

New, A. L., and Smythe-Wright, D. Aspects of the circulation in the Rockall Trough. Continental Shelf Research, 21(8-10), 777-810, https://doi.org/10.1016/S0278-4343(00)00113-8, 2001.

Teles-Machado, A., Peliz, A., McWilliams, J. C., Couvelard, X., & Ambar, I. (2016). Circulation on the Northwestern Iberian Margin: Vertical structure and seasonality of the alongshore flows. Progress in Oceanography, 140, 134-153. https://doi.org/10.1016/j.pocean.2015.05.021.

**Specific comments:**

Lines 49-51: Needs to be rephrased slightly. "… is the lack of …", "… data is scarce.", "… observations are lacking." (repetition/redundancy)

> The sentence will be rephrased to reduce redundancy related to data scarcity.

Line 52: What do you mean by "since modeling data is gridded"?

> This sentence will be rephrased to enhance clarity.

Line 103-125: Reorganize the list of sections following Figure 2: RT – CAS – WIP – ABB – MA – AS – LPP. This paragraph is also missing the Algerian-Balearic Basin (ABB) section.

> This paragraph will be adapted according to the referee's suggestions. The order of sections will be changed to follow the same order than Figure 2. It also will be added comments for ABB and 48N.

Figure 2: Provide the data set product used in Figure 2.

> The figure caption will be modified including this data.

Line 130: The Algerian Current (ALC) is missing.

> This paragraph will be modified including the missing Algerian Current.

Line 134-145: I understand that the authors define the vertical boundaries based on the T/S diagrams, but how do the authors choose the density ranges? For example, The ALC is defined as the ocean transport integrated from the surface to the 27.25 isopycnal. Based on Figure 2, this isopycnal seems to be a little bit shallow, and thus part of the ALC transport is missing.

> The authors agree that the very definition of monitoring windows entails an arbitrary component since the same current can be monitored using very different reference sections.

The selection of the region where we have chosen to monitor each current has been made based on the literature, attempting to locate the monitoring window in areas where the majority of studies describing it are concentrated and where the data we are working with seem to depict it more clearly.

However, once the region for monitoring each current has been selected, the definition of latitude and longitude values, as well as the density limits defining each current, has been established based on the mean values of the data.

This arbitrariness results in our work placing less emphasis on the absolute values of transport and more on the variability of its time series. Therefore, the transport time series (Figure 4) are expressed in terms of climatic anomalies, enabling the detection of uncertainties, trends, and relevant events while demonstrating a low dependency on the precise definition of the monitoring window.

We concur with the reviewer that, in the case of the ALC monitoring window, a vertical limit can be defined to obtain a more precise calculation of the total current transport. Thus, in the revised version of the study, we propose to adjust the vertical limit of ALC using the isopycnal of 28.5 gr/m3. In the following figure, the consequences of this change on the monitoring window are illustrated:

[Figure]

On this example the result of modifying the lower boundary of the monitoring window ALC can be observed. The isopicnal limit was modified from 27.5 gr/m3 (panels on the left column) to 28.5 gr/m3 (panels on the right column). This alteration results in a lowering of the lower limit of the monitoring window by approximately 100 m (panels in the upper row), thereby enhancing the capture of the total transport of the Algerian Current. Additionally, it is observed that the lower limit shifts from being adjacent to the main thermocline to being positioned below it (T/S diagrams in the second row).

As evident in the OMI plots presented in the third row, the outcome of this modification is practically indistinguishable. The time series exhibit a change in the obtained mean transport (0.32 Sv in the left case and 0.50 Sv in the right case). However, since the time series are presented in terms of anomalies, a modification in absolute variability is noted (anomaly values are higher in the right panel), but uncertainties, trends, and notable events remain unaltered.

Line 155: Why is the OMI aimed to focus on interannual variability? How do the mean and seasonal variabilities compare between the different products?

This OMI has been designed to compute the annual variability and medium to long-term trends of each current. This is because has significant implications such as thermohaline circulation, climate variability, and the distribution of physicochemical properties in the ocean. We agree with the reviewer that the analysis of the seasonal variability of many of the currents in question is of great interest and could have diverse environmental implications, such as fisheries management or the distribution of marine species.

However, two reasons have led us to focus exclusively on interannual variability and trends: Firstly, we believe it is preferable for the indicator to be focused on a single oceanographic aspect, simplifying the provided information and facilitating its interpretation. Secondly, given the high uncertainties obtained, we have considered that an analysis of seasonal variability could be strongly affected by these uncertainties, rendering the indicator unusable.

Lin 220: I am not sure how the journal handles the acknowledgment section. If applicable, do not forget to add the acknowledgment section (fundings, ...)

We thank for this recommendation; a short section of acknowledgements has been included.

---

## Author Response (AR2)

Dear Editor,

I am pleased to hear that the contribution has been accepted, and I would like to thank you for your assistance and cooperation. Please accept the latest version of the contribution in which the errors discovered in the last review have been corrected.

Please accept my apologies for the delay in this final iteration.

Best regards,

Álvaro de Pascual Collar